

**Long-term pig manure application increases soil organic carbon through aggregate protection**
**and Fe-carbon associations in a subtropical Red soil (Udic Ferralsols)**
Hui Rong[a,b], Zhangliu Du[c], Weida Gao[a], Lixiao Ma[d], Xinhua Peng[e], Yuji Jiang[f], Demin Yan[g], Hu Zhou[a,*]
[a] *Key Laboratory of Arable Land Conservation (North China), Ministry of Agriculture, College of Land Science and*
*Technology, China Agricultural University, Beijing, China*
[b] *State Key Laboratory of Soil and Sustainable Agriculture, Institute of Soil Sciences, Chinese Academy of Sciences,*
*71 East Beijing Road, Nanjing 210008, China*
[c] *College of Resources and Environmental Sciences, China Agricultural University, Beijing 100193, China*
[d] *State Key Laboratory of Vegetation and Environmental Change, Institute of Botany, Chinses Academy of Sciences,*
*Beijing 100093, China.*
[e] *Institute of Agricultural Resources and Regional Planning, Chinese Academy of Agricultural Sciences, Beijing*
*100081, China*
[f] *College of Resources and Environment, Fijian Agriculture and Forest University, Fuzhou, 350002, China.*
[g] *Forest Fire Research Center, Nanjing Forest Police College, Nanjing 210023, China.*
**Correspondence**: Hu Zhou (zhouhu@cau.edu.cn)
**Abstract**
Manure is known to improve soil organic carbon (SOC) in Fe-rich red soils, while the underlying
stabilization mechanisms remain poorly understood. In this study, four treatments were selected: (1)
no amendment (Control), (2) low manure (LM, 150 kg N ha$^{-1}$ yr$^{-1}$), (3) high manure (HM, 600 kg
N ha$^{-1}$ yr$^{-1}$), (4) high manure with lime (HML, 600 kg N ha$^{-1}$ yr$^{-1}$ plus 3000 kg Ca (OH)$_2$ ha$^{-1}$ 3yr$^{-1}$).
The quantity and quality of topsoil (0-20 cm) organic carbon were investigated by physical
fractionation, $^{13}$C-nuclear magnetic resonance (NMR) spectroscopy and thermogravimetry (TG)
analysis. Manure application increased total SOC by 65.1%-126.7% (primarily in the particulate
organic matter (POM) fraction), while the mineral-associated organic matter fraction (MAOM),
despite its higher C content (4.18-7.09 g C kg$^{-1}$), contributed less (65.4%-71.0%) compared to the
control (82.4%). POM C was stabilized via hierarchical aggregation: fresh manure inputs acted as
binding nuclei, increasing macroaggregates (>0.25 mm) while reducing microaggregates (0.05–0.25

* Corresponding author:
Tel number: +86 13405876151
E-mail address: zhouhu@cau.edu.cn



28 mm), physically isolating labile C from microbial decomposition. Concurrently, manure

29 amendments triggered Fe-mediated chemical stabilization. Elevated pH (4.8 to 5.4-7.1) enhanced

30 non-crystalline Fe oxide ($Fe_o$) content (+25.4%), which positively correlated with MAOM C ($R^2$ =

31 0.56, $P < 0.05$). Despite a chemical composition shift toward aliphaticity and reduced aromaticity,

32 thermally stable organic matters increased by 8%–12%, revealing critical role of $Fe_o$ (aggregates

33 were destroyed before TG analysis) in offsetting inherent molecular lability. Overall, this study

34 establishes a dual SOC stabilization framework for subtropical red soils, highlighting physical

35 protection through aggregation processes and chemical protection via Fe-carbon associations.

36 **Keywords**: Particulate organic matter; Mineral-associated organic matter; Nuclear magnetic

37 resonance; Thermogravimetry analysis

38

39 **1. Introduction**

40  Soil organic carbon (SOC), the largest carbon reservoir in the terrestrial ecosystems, plays critical

41 roles in climate mitigation and soil multifunctionality (Amelung et al., 2020; Lal, 2004). In tropical

42 and subtropical South China, red soils (Udic Ferralsols, according to Chinese Soil Taxonomy) are

43 characterized by inherently low SOC content due to intense weathering and rapid mineralization

44 (Yan et al., 2013; Zhang et al., 2013). While manure application has been widely adopted to enhance

45 SOC in these soils (Bai et al., 2023; Nichitha et al., 2023; Zhang et al., 2023), the mechanisms

46 governing SOC stabilization remain elusive. Discrepancies continue regarding whether manure

47 predominately increases SOC through chemical recalcitrance, physical protection (by aggregation

48 process), or organo-mineral interactions—given the iron-rich mineralogy of red soils and their pH-

49 dependent reactivity (Kleber et al., 2021; Six et al., 2002; Song et al., 2022).

50  Existing studies present conflicting evidence on SOC stabilization pathways. For instance,

51 Mustafa et al. (2021) reported increased aromatic C (chemically recalcitrant) with manure

52 application, whereas Yan et al. (2013) observed preferential accumulation of labile O-alkyl C. This

53 paradox highlights uncertainties in how manure inputs alter SOC composition. Fe oxides contribute

54 a lot to SOC stabilization in red soils (Zhang et al., 2013). These reactive Fe phases can form stable

55 covalent bonds between their surface hydroxyls and organic functional groups, protecting SOC from

56 microbial decomposition (Ruiz et al., 2024). In comparison with crystalline Fe oxides ($Fe_d$), non-



crystalline Fe oxides ($Fe_o$) exhibit organic matter adsorption capacity primarily attributed to their larger specific surface area (Zhang et al., 2013). The ratio of $Fe_d$ and $Fe_o$ is dynamically regulated by pH, which can be intentionally manipulated through manure application (Liu et al., 2020; Wang et al., 2023). The long-term impacts of manure-induced pH shifts on Fe oxide speciation and related organic carbon sequestration are still not well understood, despite the known connection between pH-driven Fe oxide transformation and organic matter stabilization. Additionally, previous studies often isolated chemical recalcitrance, physical protection (via aggregation), or organo-mineral interactions separately, thereby neglecting integrative assessments among these pathways. To address this knowledge gap, an integrative approach combining physical fractionation, molecular characterization, and thermal stability analysis is essential for elucidating the coupled effects of manure on SOC quantity and quality.

Physically separating soil organic matter into mineral-associated organic matter (MAOM) and particulate organic matter (POM) fractions helps predicting SOC dynamics better, and clarifying SOC stabilization mechanisms, with POM C physically protected in aggregated and MAOM C chemically protected via organo-mineral bonding (Chenu et al., 2019; Lavallee et al., 2019; Poeplau et al., 2018). While physical fractionation effectively isolates operationally defined pools (Poeplau et al., 2018), it fails to reveal chemical heterogeneity of SOC (Cotrufo et al., 2019; Lavallee et al., 2019). Solid-state $^{13}C$ nuclear magnetic resonance (NMR) spectroscopy addresses this gap by quantifying carbon functional groups (alkyl, O-alkyl, aromatic C), yet its reliance on hydrofluoric acid (HF) pretreatment risks altering native organo-mineral interactions (Kögel-Knabner, 1997). Conversely, thermogravimetry (TG) provides rapid assessment of SOC thermal stability without requiring pretreatment (Gao et al., 2015).

We hypothesized that: 1) Manure application enhance MAOM C formation by increasing non-crystalline Fe oxides ($Fe_o$), induced by elevated pH; 2) The physical protection was strengthened after manure application due to the soil aggregation process, which triggered labile SOC protection; 3) The application of pig manure strengthened the recalcitrance of SOC, thus improved thermal stability. The specific objectives of this studies were: 1) to evaluate the changes of Fe oxides and its effect on MAOM formation; 2) to explore how soil aggregation affected POM formation; 3) to evaluate the effect of manure application on SOC composition and stability.



86

## 2. Materials and methods

*2.1. Site description and experimental design*

The long-term field experiment is located at Yingtan National Agroecosystem Field Experiment Station of the Chinese Academy of Sciences (28°15′20″N, 116°55′30″E) in Jiangxi Province, China. The site has a typically subtropical humid monsoon climate with a mean annual temperature of 17.6℃ and precipitation of 1795 mm (Jiang et al., 2018). The soil is derived from Quaternary red clay, and is classified as Udic Ferralsols according to Chinese Soil Taxonomy. The soil contains 36.3% clay, 45.2% silt and 21.2% sand.

The field experiment was initiated in 2002. Four treatments were compared: (1) no manure amendment (Control), (2) low pig manure with 150 kg N ha$^{-1}$ a$^{-1}$ (LM), (3) high pig manure with 600 kg N ha$^{-1}$ a$^{-1}$ (HM), and (4) high pig manure with 600 kg N ha$^{-1}$ a$^{-1}$ and lime (HML). The four treatments received solely pig manure as the nitrogen source, with no synthetic fertilizers applied. The pig manure, collected from nearby pig farms, contained an average total carbon of 386.5 g kg$^{-1}$, total nitrogen of 36.2 g kg$^{-1}$ and total phosphorus of 21.6 g kg$^{-1}$ on a dry matter basis. The annual amount of pig manure applied to each treatment was calculated based on its nitrogen content. Since all aboveground residues (stalks and leaves) and manually recoverable roots were completely removed from the field after harvest, the total carbon inputs to the soil were derived exclusively from pig manure. This resulted in average annual carbon inputs of 1.6 and 6.4 Mg C ha$^{-1}$ for the LM and HM treatments, respectively (see Supplementary Material for calculation details). The field experiment was set up following a completely randomized design, with each treatment has three replicate plots. Each plot has a size of 2 m × 2 m. Lime was applied at 3 000 kg Ca (OH)$_2$ ha$^{-1}$ (3a)$^{-1}$ for the HML treatment. The field was planted with corn (*Zea mays L.*) monoculture annually from April to July. All the management measures, including sowing, harvesting and weeding, were manually operated.

*2.2 Sampling*

Sampling was conducted in July 2019, after the harvest of corn. Triplicate topsoil samples (0-20 cm) were randomly collected with a shovel from each plot and composited together to form one bulk sample. The soil samples were air-dried at room temperatureand were gently crushed with a



rubber mallet to pass through an 8-mm sieve, preserving aggregates >8mm for further analysis.
Visible plant residues, roots and stones were removed (Soil Survey Staff, 2011).
*2.3 Soil properties measurements*
SOC and Total nitrogen (TN) were determined by an elemental analyzer (Vario MACRO,
Elementar, Germany). Soil pH was measured using a glass electrode (PHS-3D, SANXIN, China)
with soil: deionized water ratio of 1: 2.5. Crystalline ($Fe_d$) and non-crystalline Fe oxides ($Fe_o$) were
extracted by DCB (Dithionite-citrate-bicarbonate) and oxalate, respectively (Yan et al., 2013), and
then were determined by graphite furnace atomic absorption spectrometry (GFAAS) (PinAAcle
900T, PerkinElmer, America). Water stability of aggregates was tested using the fast-wetting method
following Le Bissonnais (1996). Aggregate stability was expressed as mean weight diameter
(MWD). Detailed experimental processes and calculation can be found in Zhou et al. (2019).
*2.4. Physical fractionation*
Soil was fractionated into MAOM (<53 μm) and POM (>53 μm) fraction following Cambardella
and Elliott (1992) and Cotrofu et al. (2019). Briefly, 10 g sieved samples (<2 mm) were completely
dispersed in dilute sodium hexametaphosphate (($NaPO_3$)$_6$, 0.5%) at a soil: solution ratio of 1:4 by
shaking for 18 h (25°C, 180 r min$^{-1}$). After dispersing, soil slurry was passed through a 53 μm sieve
and rinsed several times with deionized water. The fraction passing through the sieve was collected
as MAOM fraction, and that remaining on the sieve was collected as POM fraction. Both fractions
were centrifuged and the solution was decanted, and then the remaining material was oven-dried to
constant weights at 60°C. SOC concentration of each fraction was measured using the wet oxidation
method. Dried mass proportions of each fraction (g fraction g$^{-1}$ soil) were calculated as follows:
$$f_M = m_M / m_{bulk} \qquad (1)$$
$$f_P = m_P / m_{bulk} \qquad (2)$$
where $f_M$ and $f_P$ were the dried mass proportions of MAOM and POM fraction (g fraction g$^{-1}$ bulk
soil), respectively; $m_M$ and $m_P$ (g) were the dried masses of MAOM and POM fractions; $m_{bulk}$ (g)
was the dried mass of bulk soil.
SOC in the MAOM and POM fractions were called as MAOM C and POM C, respectively in
this paper. MAOM C and POM C were calculated by multiplying the dried mass proportions of each
fraction (g fraction g$^{-1}$ soil) by the respective SOC concentrations (g C kg$^{-1}$ fraction) as follows



(Garten and Wullschleger, 2000; Lian et al., 2015):
$$\text{MAOM C} = f_M \times \text{SOC}_M \qquad (3)$$
$$\text{POM C} = f_P \times \text{SOC}_P \qquad (4)$$
where $f_M$ and $f_P$ were calculated by Equation (1) and Equation (2); $\text{SOC}_M$ and $\text{SOC}_P$ were the SOC
concentrations in the MAOM and POM fraction (g C kg$^{-1}$ fraction), respectively.
The contributions of MAOM C and POM C to total SOC (%) was calculated as:
$$\text{Contribution of MAOM C (\%)} = \text{MAOM C} / \text{Total SOC} \times 100 \quad (5)$$
$$\text{Contribution of POM C (\%)} = \text{POM C} / \text{Total SOC} \times 100 \qquad (6)$$
where MAOM C and POM C were derived from Equations (3) and (4), respectively; total SOC was
the content of SOC in the bulk soil.
*2.5. SOC chemical composition and chemical stability*
SOC chemical composition was analyzed with a solid-state cross-polarization magic angle
spinning (CPMAS) $^{13}$C nuclear magnetic resonance (NMR) spectroscopy. Prior to NMR analysis,
<2 mm air-dried soils were pretreated with hydrofluoric acid (HF) to remove paramagnetic $Fe^{3+}$
(iron oxides) following Gao et al. (2021). Firstly, 20 g soil was mixed with 100 mL 10% (w/w) HF
solution in a polyethylene bottle and then shaken for 0.5 h per day for three days. Afterwards, the
supernatant liquid was discarded and another 100 mL 10% HF solution was added again. The above
procedures were repeated for 15 times. The residue was rinsed 10 times with deionized water until
the pH was close to neutral. The remaining soil was freeze-dried and then ground in an agate mortar
to pass through a 100-mesh sieve (0.149 mm) for further analysis. This fine grinding ensured
homogeneous packing in the NMR rotor, minimizing signal heterogeneity (Simpson & Simpson,

165     2012).

Carbon functional groups were determined with the Bruker Ascend 500 MHz NMR spectrometer
(Bruker BioSpin, Rheinstetten, Germany). Dry powdered samples were placed in a 4-mm sample
rotor operating at a $^{13}$C resonance frequency of 125.8 MHz. The NMR spectrometer run at a spinning
rate of 5kHz, and 10500 scans were collected for each sample. The spectra were collected over an
acquisition time of 12 ms and a recycle delay of 0.8 s. We assigned the obtained spectra to four
different carbon functional groups, i.e., alkyl C (0-45 ppm), O-alkyl C (45-110 ppm), aromatic C
(110-160 ppm) and carbonyl C (160-220 ppm) according to Kögel-Kanbner (1997). The relative



concentrations of the different functional groups were calculated as the percentage of their peak
areas to the total aeras using MestReNova 14.0 software (Mestrelab Research, 2019, Spain). Indices
used to evaluate SOC recalcitrance include: alkyl C/O-alkyl C, aromaticity, aromatic C/O-alkyl C,
aliphatic C/aromatic C and aliphaticity (Baldock et al., 1997; Du et al., 2017). Aromaticity =
aromatic C / (alkyl C + O-alkyl C + aromatic C); Aliphatic C = alkyl C + O-alkyl C; Aliphaticity =
(alkyl C + O-alkyl C) / (alkyl C + O-alkyl C + aromatic C). The alkyl C/O-alkyl C ratio reflects the
degree of microbial transformation, with higher values indicating advanced decomposition and
accumulation of recalcitrant alkyl compounds (Baldock et al., 1997). The aromatic C/O-alkyl C
ration aligns with the alkyl C/O-alkyl C ration in reflecting the degree of SOC decomposition,
whereas the aliphatic C/aromatic C ratio presents a contrary perspective to them. Aromaticity is a
chemical concept denoting the resistance to microbial degradation (Kögel-Knabner, 1997).
Aliphaticity is used to quantify the proportion of labile aliphatic components relative to stable
aromatic moieties.
*2.6. Thermogravimetry analysis*
TG analysis was performed using a Netzsch TG 209F1 (Netzsch-Gerätebau GmbH, Selb,
Germany). Air-dried soil samples were first sieved through a 2-mm mesh, and then ground in a ball
mill to pass through a 50 μm sieve. The grounded soil samples (5 ~ 10 mg) were placed in an $Al_2O_3$
crucible covered with an aluminum lid and were oxidized in an atmosphere of 20 mL min$^{-1}$ of
synthetic air (20% $O_2$ and 80% $N_2$) and 20 mL min$^{-1}$ of $N_2$ as a protective gas. The temperature
program included a heating rate of 10$^{\circ}$C min$^{-1}$ from 40$^{\circ}$C up to 800$^{\circ}$C. The sample mass percentage
relative to the initial mass as a function of temperature was recorded simultaneously, and its first
derivative (DTG) was calculated to represent the mass loss rate.
Three processes were detected in the temperature range of 40$^{\circ}$C to 800$^{\circ}$C: hygroscopic moisture
evaporation, SOM decomposition and carbonate breaking down (Gao et al., 2015; Siewert, 2004).
Based on the observed DTG curve, the weight loss between 180$^{\circ}$C and 530$^{\circ}$C, representing the
temperature range during which SOM was decomposed, was defined as the total mass of SOM
($Exo_{tot}$). The mass loss between 180$^{\circ}$C and 380$^{\circ}$C was a consequence of thermally labile SOM
oxidation ($Exo_1$) and that between 380$^{\circ}$C and 530$^{\circ}$C was caused by the combustion of more stable
organic matter ($Exo_2$) (Gao et al., 2015; Volkov et al., 2020). We used two parameters, the ratio of



$Exo_1$ and $Exo_{tot}$ ($Exo_1$/ $Exo_{tot}$) and the temperature at which half of the SOM was decomposed (TG-
$T_{50}$), to characterize the thermal stability of SOM. Higher $Exo_1$/ $Exo_{tot}$ and lower TG-$T_{50}$ values
indicate that the sample has more thermally labile or unstable SOM (Gao et al., 2015; Siewert, 2004).
*2.7. Statistical analysis*
Statistical analyses were carried out with R Studio software (R Development Core Team, version
4.1.2). One-way analysis of variance (ANOVA) was conducted to assess the effect of amendments
on soil physico-chemical properties, SOC physical fractions, chemical composition and thermal
indices. The independence of samples, normality of residues and homogeneity of variances were
checked by Chisq, Shapiro-Wilk and Bartlett test, respectively. Fisher's least significant difference
(LSD) method was used for the multiple comparisons of means with a 0.05 significance level. Linear
regression analysis was conducted to investigate the correlations between SOC and iron oxides, soil
aggregation. Principal component analysis (PCA) was performed to evaluate the relationship
between the quantity and quality of SOC and factors related to chemical protection and physical
protection. Pearson correlation analyses were performed to explore the relationships between
chemical composition and thermal indices.
**3. Results**
*3.1. Long-term pig manure application increased SOC, TN, pH, non-crystalline ($Fe_o$) and*
*improved soil aggregation*
Long-term manure amendment altered the soil physic-chemical properties (Table 1). Relative to
Control, the LM, HM and HML treatments increased SOC concentration by 64.9%, 116.2% and
126.6%, respectively ($P < 0.05$), and increased TN concentration by 48.0%-108.2% ($P < 0.05$). The
pH values were increased by 0.62-2.28 units after pig manure application ($P < 0.05$). Application of
pig manure had no significant effect on crystalline iron oxides ($Fe_d$) content ($P > 0.05$), but the HM
and HML treatments significantly increased non-crystalline iron oxides ($Fe_o$) by 25.4% ($P < 0.05$).
Relative to Control, the HM and HML treatments significantly increased macroaggregates (>0.25
mm) content by 15.8% and 16.8%, respectively, and they increased MWD by 24.3% and 35.0%,
respectively ($P < 0.05$). Microaggregates (0.05-0.25 mm) was decreased by 30.4% and 36.4%,
respectively under the HM and HML treatments ($P < 0.05$).
*3.2. Long-term pig manure application affected SOM physical fractions: MAOM and POM*



The distribution of MAOM and POM fractions was significantly influenced by manure and lime
amendments ($P < 0.05$, Table 2). Across all treatments, MAOM dominated the soil mass proportion
(72.5%–75.1%), whereas POM mass proportion increased progressively from 16.6% in the control
to 19.3%–19.4% under the HM and HML treatments.
Manure application improved SOC concentration in both fractions, with increase rates of 32.0%-
66.8% and 208%-592% in the MAOM and POM fractions, respectively. Despite this,
the contribution of MAOM to total SOC declined from 82.4% in the control to 65.5%–65.8% under
the HM and HML treatments, while POM contribution increased nearly threefold (from 8.8% to
23.7%–26.0%). Lime addition (HML vs. HM) did not significantly alter mass proportions but
further enhanced POM C concentration (+15.4%) and its SOC contribution (+9.7%).
**3.3. Effect of long-term pig manure application on SOC chemical composition and recalcitrance**
The solid-state $^{13}$C NMR spectra showed different signal patterns for the different treatments (Fig.
1) and quantified the ratios of the different SOC functional groups shown in Table 3. Relative to
Control, the HM and HML treatments significantly increased alkyl C by 4.6%-4.9% ($P < 0.05$),
while the LM treatment showed no significant change ($P > 0.05$). The O-alkyl C was significantly
increased by 5.5%, 2.6% and 2.2% under the LM, HM and HML treatments, respectively ($P < 0.05$).
Aromatic C and carbonyl C were decreased by 12.4%-13.2% and 0.9%-8.7%, respectively, in the
manured treatment ($P < 0.05$). While aromatic C was increased in the content after manure amend,
its relative proportion was decreased by 12.4%-13.2% ($P < 0.05$).
Relative to Control, the LM treatment significantly decreased alkyl C/O-alkyl C ratio ($P < 0.05$),
but the HM and HML treatments had no significant effect on the ratio ($P > 0.05$, Table3). The
aromaticity was decreased by 14.4%, 12.9% and 13.2% under LM, HM and HML treatments,
respectively ($P < 0.05$, Table3). Similarly, the aromatic C/O-alkyl C ratio was decreased by 14.7%-
17.7% in the manured treatment ($P < 0.05$, Table3). In contrast, the aliphatic C/aromatic C ratio and
aliphaticity were increased by 18.1%-20.7% and 2.44%-3.66% in the manured treatments,
respectively ($P < 0.05$, Table3).
**3.4. The effect of long-term pig manure application on SOC thermal stability**
The shapes of TG and its first derivatives (DTG) curves showed distinct weight losses rate as
temperature increased to above 100℃, in the order of HM>HML>LM>Control (Fig. 2). Relative to



Control treatment, pig manure application significantly increased the total mass losses in the range of 180°C to 530°C ($Exo_{tot}$) by 11.1%- 17.4% ($P < 0.05$, Table 4), and significantly increased the mass losses during 180-380°C ($Exo_1$) and 380-530°C ($Exo_2$) by 14.6%-26.5% and 7.8%-13.7%, respectively ($P < 0.05$, Table 4). The LM and HML treatments significantly increased the ratio of $Exo_1/Exo_{tot}$ ($P < 0.05$), whereas the HM treatment had no significant effect ($P > 0.05$, Table 4). HM and HML treatments significantly decreased TG-$T_{50}$ by 10.7-12.0 °C ($P < 0.05$), while LM treatments showed no significant difference compared to Control ($P > 0.05$, Table 4).

**3.5. *Relationships between SOC and factors related to the mineral protection and physical protection of SOC***

Fig. 3 showed correlations between SOC and possible variables associated with the chemical protection and physical protection of SOC. SOC in the bulk soil was significantly positively correlated with MAOM C ($R^2$=0.97, Fig. 3A). While MAOM C showed no relationship with $Fe_d$ ($R^2$=0.04, Fig. 3B), it was positively correlated with $Fe_o$ ($R^2$=0.56, Fig. 3C). In addition, there was no correlation between MAOM C and the content of clay and silt ($R^2$=0.34, Fig.3D).

SOC in the bulk soil was significantly positively correlated with POM C ($R^2$=0.95, Fig. 3E). POM C was significantly associated with soil aggregation, evidenced by the positive correlation with macroaggregates (>0.25 mm) ($R^2$=0.78, Fig. 3F) and MWD ($R^2$=0.67, Fig. 3H), while negative correlation with microaggregates (<0.25 mm) ($R^2$=-0.71, Fig. 3G).

A PCA plot diagram (Fig. 4) revealed distinct associations between SOC quantity/quality and stabilization mechanisms. SOC vector aligned positively with POM C, macroaggregates and MWD, but inversely with microaggregates. Chemically, SOC covaried with MAOM C and $Fe_o$, yet formed an obtuse angle (>90°) with $Fe_d$. TG-$T_{50}$ negatively associated with O-alkyl C and aliphaticity (angles > 90°), but positively linked to aromatic C and aromaticity (angles < 90°).

**4. Discussion**

***4.1. POM C and physical protection***

Long-term pig manure application caused an increase of SOC in the bulk soil, and the increase was mainly derived from continuous manure inputs (Gong et al., 2009). Although maize rhizodeposition (including root exudates and sloughed-off cells) (typically <10% of net primary productivity; Pausch and Kuzyakov, 2018) contributes to SOC during the growing season, its annual



carbon inputs is only 0.2-0.5 Mg C ha$^{-1}$ yr$^{-1}$ (Dennis et al., 2010). This contribution is negligible in
comparison with manure inputs, which ranged from 1.6 to 6.4 Mg C ha$^{-1}$ yr$^{-1}$ in the LM and HM
treatments, respectively. Furthermore, the rigorous removal of all harvest residues (stalks and
recoverable roots) excluded aboveground and root biomass as a source of soil carbon deposition.
Thus, the SOC increase in the LM/HM treatments (Table 1) can be predominately attributed to the
exogenous organic C supplied by pig manure, and the HML treatment can be partly influenced by
lime. Following microbial decomposition and transformation, the applied manure-C was
progressively partitioned into distinct SOC pools.
Manure-derived carbon was preferentially allocated to the POM fraction rather than the MAOM
fraction. The contribution of POM C to SOC increased nearly threefold (8.8% to 26.0%), while that
of MAOM C decreased significantly from 82.4% in the Control to 65.4%–71.0% under manure
application treatments (Table 2). The result was in line with Lan et al. (2022) and Li et al. (2018),
who reported that increased manure substitution improved the POM C/MAOM C ratio, indicating
manure was beneficial to the formation of POM C. Recent biomarker evidence demonstrated that
plant-derived carbon contributes disproportionately to POM in comparison with MAOM fraction
(Zou et al., 2023). Notably, $^{13}$C isotopic tracing study revealed that 70%-87% of residue-derived
SOC was accumulated in the POM fraction at two sites in the study of Mitchell et al. (2021). These
results jointly validate that fresh organic inputs are distributed primarily in the POM fraction due to
direct occlusion within macroaggregates before microbial processing. Therefore, POM C can be a
good indicator of SOC dynamics under field management (Álvaro-Fuentes et al., 2021; Wu et al.,

309    2023).

POM C was susceptible to decomposition due to its rapid dynamics, whereas it was
simultaneously protected through physical occultation within soil aggregates. According to the
classical hierarchical aggregation model (Six et al., 2000), organic carbon acts as a primary binding
agent for microaggregates (<0.25mm) to form macroaggregates (>0.25mm), with POM serving as
both a structural nucleus and a transient carbon reservoir (Six et al., 2000). In this study,
macroaggregates were significantly increased, while microaggregates were significantly decreased
after pig manure application (Table 1). These results implied that manure-derived carbon bound
microaggregates to form macroaggregates, thus providing physical protection for SOC in the POM



fraction (Peng et al., 2023). The observed positive correlation between POM C and both
macroaggregates and aggregate MWD (Fig.3; Fig.4) further verified the critical role of soil
aggregation. However, the physical mechanisms by which POM facilitates this process (e.g.,
microbial mediation, hydrophobic interactions, or polysaccharide bridging) require further
investigation. Experimental approaches such as isotopic labelling combined with micro-scale
imaging (e.g., electron microscopy or X-ray computed tomography) can visualize the spatial
distribution of POM within aggregates and quantify its role in aggregate formation. Future study
should pay attention to these new technologies.

### 4.2. MAOM C and chemical protection

MAOM retained higher SOC concentrations (4.18-7.09 g C $kg^{-1}$ bulk soil) irrespective of
treatments, though its contribution to total SOC decreased after manure application (Table 2).
MAOM C exhibited significantly longer mean residence time (MRT) compared to POM C, with
reported turnover periods of 26-40 years versus 2.4-4.3 years, respectively (Benbi et al., 2014;
Garten and Wullschleger, 2000). This fundamental difference originates from the unique
stabilization mechanism of MAOM C through persistent organo-mineral associations (Lavallee et
al., 2019). While clay minerals are widely recognized as key MAOM stabilizers (Hemingway et al.,
2019; Liang et al., 2017), our study reveals a distinct iron oxide-dominated mechanism in these Fe-
rich red soils. The strong correlation between MAOM C and non-crystalline Fe oxides ($Fe_o$, $R^2$=0.56;
Figs. 3C, 4) contrasts with its weak relationship with clay + silt content ($R^2$=0.34; Fig. 3D),
highlighting the pivotal role of $Fe_o$ in this red soil. This iron-mediated stabilization likely stems from
the exceptionally high specific surface area of $Fe_o$ (~800 $m^2$ $g^{-1}$ in ferrihydrite; Kleber et al., 2005)
and its superior capacity to form stable organo-mineral complexes (Eusterhues et al., 2005;
Lehmann and Kleber, 2015). The increased pH after manure application (Table 1) created conditions
favoring $Fe_o$ preservation (Vithana et al., 2015), with an increase of 25.4% in the content of $Fe_o$.
The increased $Fe_o$ content provided abundant reactive surfaces for MAOM formation. This pH-$Fe_o$-
MAOM nexus establishes a self-reinforcing stabilization mechanism: manure-derived organic
ligands interact with $Fe_o$ to form stable complexes, simultaneously protecting both organic carbon
and $Fe_o$ from dissolution (Kleber et al., 2021).





The positive correlation between $Fe_o$ and MAOM C (Fig. 3C) suggested that non-crystalline Fe
oxides played a critical role in stabilizing SOC through organo-mineral interactions. However, while
our data support the association between $Fe_o$ and MAOM C, the underlying mechanisms (e.g.,
adsorption, co-precipitation, or ligand exchange) remain speculative due to the lack of direct
molecular-scale evidence. Future studies employing advanced spectroscopic techniques (e.g.,
synchrotron-based X-ray absorption spectroscopy or NanoSIMS) could explicitly characterize the
binding forms of Fe-organic complexes, thereby validating the causal relationship between Feo and
MAOM formation.
***4.3. Chemical composition and thermal stability***
Input of new organic material could alter chemical composition of SOC and lead to the change
of the molecule recalcitrance of SOC (Guo et al.,2019; Yan et al., 2013; Zhou et al., 2010; Zhang et
al., 2013). In the current study, pig manure application increased the content of O-alkyl C, but
declined that of aromatic C. Pig manure was rich in cellulose and lignin components, so the
introduction of manure greatly increased O-alkyl C (Li et al., 2015). The considerable increase of
O-alkyl C accounted for the decrease of the relative proportion of aromatic C and the decrease of
aromaticity.
Long-term pig manure application strengthened SOC thermal stability by improving the content
of thermally stable organic matters while it decreased TG-$T_{50}$. Thermal analysis suggested that
manure amend significantly increased SOM content, evidenced by the increase of the mass losses
during 180-530 ℃ ($Exo_{tot}$) (Table 4). The result was consistent with the change of SOC content
measured by conventional method (Table 1), indicating TG technology was promising for
measuring SOM content (Siewert, 2004; Tokarski et al., 2018). The decrease of TG-$T_{50}$ reflected
more easily decomposable SOC accumulated after manure application (Gao et al., 2015; Siewert,
2004). The result consisted with that in the NMR spectroscopy, where greater O-alkyl C and higher
aliphaticity were found under manure treatments (Table 3). SOC with more O-alkyl C functional
groups or higher aliphaticity was less likely to resist thermochemical degradation as revealed by the
negative relationship between TG-$T_{50}$ and aliphacity (Fig.4; Fig. 5) (Hou et al., 2019; Lehman and
Kleber, 2015), thus leading to a decrease of TG-$T_{50}$. In this study, the thermally labile organic
matters accounted for over half of the total organic matters, so the decrease of TG-$T_{50}$ after manure



application was just a result of the increased thermally organic matters not the decrease of thermal
stability. In contrast, thermal stability should be strengthened according to the increased thermally
stable organic matters after manure application. Since soil structure was destroyed due to the ground
process of soil samples before TG analysis, the increase of the thermally stable organic matters was
ascribed to mineral protection, where the correlations between organic carbon and Fe oxides
increased the thermally resistance of OC (Ruiz et al., 2023).

**5. Conclusion**

Manure application increased SOC quantity and improved its quality in Fe-rich red soils. SOC in
the POM fraction exhibited the most pronounced response to manure inputs, while the majority of
SOC was stored in the MAOM fraction. Furthermore, SOC was stabilized by distinct yet
complementary mechanisms: physical protection via aggregation process, and chemical protection
via Fe-organic associations induced by elevated pH. In addition, manure application increased
thermally stable organic matters. However, to better understand inherent mechanisms, future work
should focus on molecular-scale characterization of Fe-organic interactions using synchrotron
techniques (e.g., Fe K-edge XANES/EXAFS), and in-situ visualization of POM-mediated
aggregation through advanced imaging tools (e.g., SEM-TEM or μ-CT) coupled with $^{13}$C-labelled
manure to track POM dynamics within aggregates.

**Acknowledgments**

This work was financially supported by the NSFC-CAS Joint Fund Utilizing Large-scale Scientific
Facilities (No. U1832188) and Chinese Universities Scientific Fund (2023RC047). We thank
Yanyan Cai and Xing Xia for laboratory assistance.

**Author contributions**

HR conceived the experimental approach, took soil samples from the field, carried out the laboratory
and data analyses, wrote the first draft of the manuscript, and contributed to subsequent drafts. ZLD
helped analyze NMR spectrum and revise the manuscript. WDG contributed to interpretating TG
data and revising the manuscript. LXM conducted the experiment of NMR. XHP helped design the
experiment, and analyze data. YJJ contributed to the design of field experiment and the process of



taking soil samples. DMY contributed to the process of writing. HZ contributed to funding

acquisition, conceiving the experimental approach, carrying out data interpretations, and the writing

of all subsequent manuscript drafts.

**Conflict of Interest**

Hu Zhou is a member of the editorial board of SOIL.

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



**Table 1**

Soil organic carbon (SOC), total nitrogen (TN), the ration of SOC to TN (SOC/TN), pH, crystalline iron oxides ($Fe_d$), non-crystalline iron oxides ($Fe_o$), aggregate size distribution and mean weight diameter (MWD) of water-stable aggregates under no manure (Control), low manure (LM), high manure (HM) and high manure plus lime (HML) treatments.

| Treatments | SOC (g kg⁻¹) | TN (g kg⁻¹) | SOC/TN | pH | Fe_d (g kg⁻¹) | Fe_o (g kg⁻¹) | Aggregate size distribution (g g⁻¹) | | | MWD (mm) |
|---|---|---|---|---|---|---|---|---|---|---|
| | | | | | | | >0.25 mm | 0.05-0.25 mm | <0.05 mm | |
| Control | 4.79c | 0.67c | 7.10b | 4.80c | 62.99a | 1.81b | 0.67c | 0.28a | 0.05a | 0.98b |
| LM | 7.91b | 1.00b | 7.94a | 5.42c | 59.98a | 1.74b | 0.71b | 0.25a | 0.04ab | 1.09b |
| HM | 10.37a | 1.35a | 7.69ab | 6.11b | 59.59a | 2.25a | 0.77a | 0.19b | 0.03b | 1.22a |
| HML | 10.86a | 1.40a | 7.76ab | 7.08a | 62.22a | 2.29a | 0.78a | 0.18b | 0.04ab | 1.32a |

Values are the means (n=3). Different lowercase letters after values in the same row indicate a significant difference among four manure treatments ($P < 0.05$).

**Table 2**

Mass proportion, SOC concentration and contribution of the mineral-associated organic matter (MAOM) (<53 μm) fraction, particulate organic matter (POM) (>53 μm) fraction under no manure (Control), low manure (LM), high manure (HM) and high manure plus lime (HML) treatments.

| Treatments | Mass proportion (%) | | SOC concentration (g C kg⁻¹ soil) | | Contribution to total SOC/% | |
|---|---|---|---|---|---|---|
| | MAOM | POM | MAOM | POM | MAOM | POM |
| Control | 74.3a | 16.6b | 4.18c | 0.44c | 82.4a | 8.8c |
| LM | 75.1a | 17.2ab | 5.61b | 1.26b | 71.0b | 15.9b |
| HM | 73.9a | 19.4a | 6.81a | 2.46a | 65.8b | 23.7a |
| HML | 72.5a | 19.3a | 7.09a | 2.84a | 65.4b | 26.0a |

Values are the means (n=3). Different lowercase letters after values in the same row indicate a significant difference among four manure treatments ($P < 0.05$). Calculations follow the methodology described in Section 2.4.





**Table 3**
The contents of various C functional groups in CPMAS-$^{13}$C-NMR spectra under no manure (Control), low manure
(LM), high manure (HM) and high manure plus lime (HML) treatments. Alkyl (0-45 ppm); O-alkyl C (45-110
ppm); aromatic C (110-160 ppm) and carbonyl C (160-220 ppm).

| Treatments | alkyl C (%) | O-alkyl C (%) | aromatic C (%) | carbonyl C (%) | alkyl C/ O-alkyl C | aromaticity (%) | aromatic C/O- alkyl C | aliphatic C/ aromatic C | aliphaticity (%) |
|---|---|---|---|---|---|---|---|---|---|
| Control | 25.4b | 44.9c | 15.3a | 14.4a | 0.57a | 17.9a | 0.34a | 4.58b | 82.1b |
| LM | 26.2ab | 47.4a | 13.3b | 13.1b | 0.55b | 15.3b | 0.28b | 5.53a | 84.7a |
| HM | 26.6a | 46.1b | 13.4b | 13.9ab | 0.58a | 15.6b | 0.29b | 5.41a | 84.4a |
| HML | 26.5a | 45.9b | 13.3b | 14.3a | 0.58a | 15.6b | 0.29b | 5.43a | 84.4a |

Values are the means (n=3). Different lowercase letters after values in the same row indicate a significant difference
among four manure treatments ($P < 0.05$). Aromaticity = aromatic C/ (alkyl C + O-alkyl C + aromatic C); aliphatic
C = (alkyl C + O-alkyl C); aliphaticity = (alkyl C + O-alkyl C) / (alkyl C + O-alkyl C + aromatic C).

**Table 4**
The mass losses during specific temperature ranges under no manure (Control), low manure (LM), high manure
(HM) and high manure plus lime (HML) treatments. Exo$_1$ represents thermally labile soil organic matter (SOM);
Exo$_2$ represents thermally stable SOM; Exo$_{tot}$ represents total SOM. TG-T$_{50}$ indicates the temperature at which half
of the total SOM is lost.

| Treatments | Exo$_1$ (180~380℃) (%) | Exo$_2$ (380~530℃) (%) | Exo$_{tot}$ (180~530℃) (%) | Exo$_1$/Exo$_{tot}$ | TG-T$_{50}$ (℃) |
|---|---|---|---|---|---|
| Control | 2.28c | 2.36b | 4.64b | 0.49b | 381a |
| LM | 2.62b | 2.54a | 5.16a | 0.51a | 376ab |
| HM | 2.69ab | 2.68a | 5.37a | 0.50b | 369b |
| HML | 2.89a | 2.56a | 5.45a | 0.53a | 371b |

Values are the means (n=3). Different lowercase letters after values in the same row indicate a significant difference
among four manure treatments ($P < 0.05$).














**Figure legends**

**Fig. 1** CPMAS-$^{13}$C-NMR spectra under no manure (Control), low manure (LM), high manure (HM) and high
manure plus lime (HML) treatments. Alkyl (0-45 ppm); O-alkyl C (45-110 ppm); aromatic C (110-160 ppm) and
carbonyl C (160-220 ppm)

**Fig. 2** Thermogravimetry (TG) curves (a) and corresponding derivative thermogravimetry (DTG) curves (b) of soil
organic matter (SOM) under no manure (Control), low manure (LM), high manure (HM) and high manure plus
lime (HML) treatments

**Fig. 3** The relationships between SOC and variables related to the chemical protection and physical protection

**Fig. 4** Biplots of the principal component analysis (PCA) between the quantity and quality of SOC and variables
related to chemical protection, physical protection across four manure application treatments (CK, Control; LM,
low manure; HM, high manure; and HML, high manure plus lime)

**Fig. 5** The relationships between SOC quantity, chemical recalcitrance and thermal stability




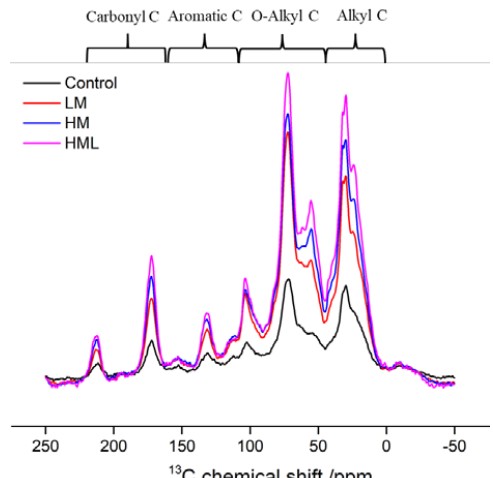


**Fig.1** CPMAS-¹³C-NMR spectra under no manure (Control), low manure (LM), high manure (HM) and high
manure plus lime (HML) treatments. Alkyl (0-45 ppm); O-alkyl C (45-110 ppm); aromatic C (110-160 ppm) and
carbonyl C (160-220 ppm)





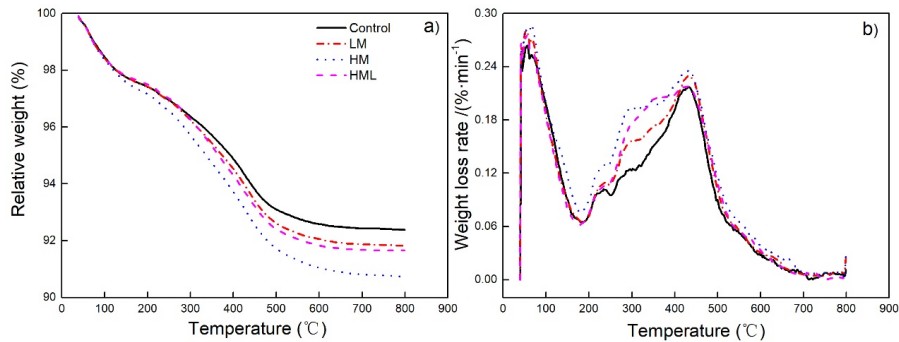

**Fig. 2** Thermogravimetry (TG) curves (a) and corresponding derivative thermogravimetry (DTG) curves (b) of soil organic matter (SOM) under no manure (Control), low manure (LM), high manure (HM) and high manure plus lime (HML) treatments



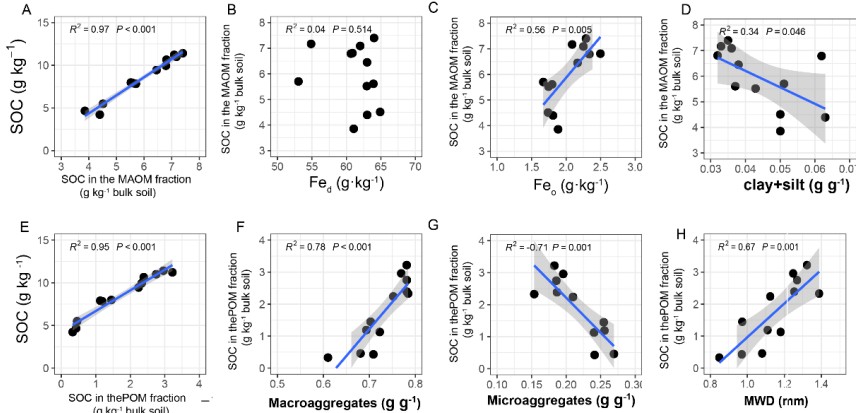

**Fig. 3** The relationships between SOC and variables related to the chemical protection and physical protection. (A to D) Variables characterizing chemical protection, including the content of SOC stored in the MAOM fraction, the content of crystalline Fe oxides ($Fe_d$), the content of non-crystalline Fe oxides ($Fe_o$), and the content of clay and silt. (E to H) Variables characterizing physical protection, including the content of SOC stored in the POM fractions, the content of soil macroaggregates, the content of soil microaggregates and the mean weight diameter (MWD)

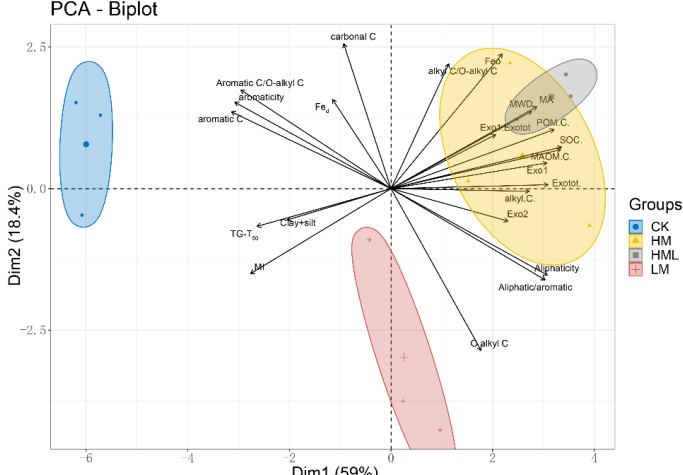

**Fig. 4** Biplots of the principal component analysis (PCA) between the quantity and quality of SOC and variables related to chemical protection, physical protection across four manure application treatments (CK, Control; LM, low manure; HM, high manure; and HML, high manure plus lime). $Fe_d$, crystalline Fe oxides; $Fe_o$, non-crystalline Fe oxides; MA, macroaggregates (>0.25 mm); MI, microaggregates (0.05-0.25 mm); MWD, mean weight diameter; aromaticity, aromatic C / (alkyl C + O-alkyl C + aromatic C); aliphatic C, Alkyl C + O-alkyl C; aliphaticity, (Alkyl C + O-alkyl C) / (Alkyl C + O-alkyl C + Aromatic C); $Exo_1$, thermally labile soil organic matter (SOM); $Exo_2$, thermally stable SOM; $Exo_{tot}$, total SOM; TG-$T_{50}$, the temperature at which half of the total SOM was lost



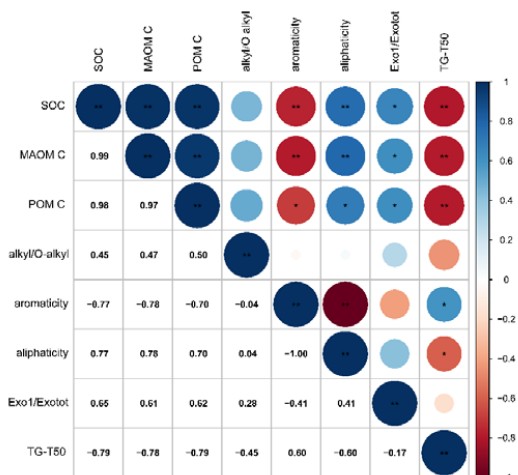

666

**Fig. 5** The relationships between SOC quantity, chemical recalcitrance and thermal stability. Aromaticity, aromatic C / (alkyl C + O-alkyl C + aromatic C); aliphaticity, (alkyl C + O-alkyl C) / (alkyl C + O-alkyl C + aromatic C); $Exo_1$, thermally labile soil organic matter (SOM); $Exo_{tot}$, total SOM; $TG\text{-}T_{50}$, the temperature at which half of the total SOM was lost