# Peer review of "Long-term pig manure application increases SOC through aggregate protection and Fe-C"

_EGUsphere, 2025_

## Author Comment (AC1)

*Lines 722-724:*

[Figure]

**Fig. 4** *Relationship between soil pH and iron oxides concentrations. (A) Significant positive correlation between pH and amorphous iron oxides (Fe$_o$). (B) No significant correlation between pH and crystalline iron oxides (Fe$_d$).*

*Lines 368-373: "Previous studies have demonstrated that organic amendments can enhance the content of non-crystalline Fe oxides, thereby promoting Fe-C associations (Chen et al., 2022; Huang et al., 2017; Wang et al., 2019). Extending these findings, our results provide new mechanistic insight by demonstrating that the increase in non-crystalline Fe oxides is closely linked to manure-induced soil alkalization. Specifically, the significant positive correlation between pH and Fe$_o$ indicates that the…"*

*Line701-702:*

[Figure]

**Fig.1** *CPMAS-$^{13}$C-NMR spectra under no manure (Control), low manure (LM), high manure (HM) and high manure plus lime (HML) treatments. Alkyl (0-45 ppm); O-alkyl C (45-110 ppm); aromatic C (110-160 ppm) and carbonyl C (160-220 ppm). The spectra are normalized to the total integral area of the control spectrum to highlight differences in the relative abundances of carbon functional groups.*

---

## Author Response (AR1)

Dear Editor,

Thank you for your comments which are very helpful to improve our manuscript. We have carefully considered all your suggestions and have incorporated them into the revised version. All modifications in the revision are highlighted in yellow. We believe these revisions have greatly improved the paper and are grateful for your help.

**General comments:**

- **More information on the study design is required to aid interpretation**. Please clarify the precise timing of the treatment initiation and manure application, and for how long the field had been planted with corn prior to the start of the experiment. What are the general practices on the field other than manure application, e.g. was the field tilled at any time? Please describe whether manure was incorporated into the soil or left on the surface, and if possible, its moisture content at the time of application and whether there was any visible manure remaining at sampling.

**Response**: Detailed information on the study design is provided in the revision and they are listed below. Please find our detailed responses below:

The experiment was initiated in April 2002. The soils used in this experiment were transported from adjacent forestland and repacked (to a depth of 2 m) for all the plots. This design ensures a uniform initial soil condition and eliminates the influence of prior land use history. We have added the text in lines 112-114.

Pig manure is applied annually at April, prior to corn sowing. The pig manure was applied onto the soil surface (0–10 cm depth) and subsequently incorporated into the topsoil through plowing and harrowing (lines 128-130).

The moisture content of the fresh manure was not measured at the time of application. At the time of sampling (July, approximately three months after application), no visible manure residues were observed (lines 137-138), which is consistent with the expected decomposition rate under warm and humid summer conditions.

The clarifications have also been added to the revised manuscript in the Materials and Methods section (lines112-114, lines 128-130 and lines 137-138).

*Lines 112-114: "The field experiment was initiated in April, 2002. The soils used in this experiment were transported from adjacent forestland and repacked (to a depth of 2 m) for all the plots. This design ensures a uniform initial soil condition and eliminates the influence of prior land use history."*

*Lines 128-130: "...The pig manure was applied annually each April onto the soil surface (0-10 cm depth) and*

*subsequently incorporated into the topsoil through plowing and harrowing..."*

*Lines 137-138: "...No visible manure residues were observed."*

**- Measurements were only made at one time, and all of the comparisons are assuming a space-for-time substitution.** Terms such as "increase" and "decrease" are misleading, especially if the space-for-time assumption is not clearly stated before using such terms and the associated risks of misinterpretation (e.g., that differences do not represent true change if the starting point was not identical) is not represented throughout the manuscript. I suggest using terms such as "higher" and "lower" "relative to the control" when describing the results to avoid misleading the reader.

**Response**: We agree and revised the manuscript as suggested. that the use of terms such as "increase" and "decrease" can be misleading in a space-for-time substitution design without clear clarification. We replaced dynamic terms like "increase" and "decrease" with more accurate descriptive terms such as "higher in", "lower in", "greater than", and "relative to the control" to describe the differences among treatment. Due to the extensive revisions made throughout the manuscript, we have incorporated all changes directly into the text and highlighted them in yellow rather than listing each one here for the sake of brevity.

**Specific comments:**

**L101-104: I do not think that total C inputs to the soil can be "exclusively" from manure (perhaps "mainly")**, as C inputs from plant tissue turnover and root exudation while the corn is growing are impossible to avoid. For example, plant root material could have been incorporated into aggregates during the growing season (and not removed at harvest or during soil processing) thereby contributing significantly to either POC or MAOC. Further, L 116 implies that visible plant residues and roots remained in the soil post sampling. This is acknowledged somewhat in the discussion (L286-292) but the current word choice both here and in the discussion is misleading. Please soften the terms used to communicate that some C is coming from plants (tissues and exudation), and given it was not directly measured, it can't be said whether or not the amounts were completely negligible. (Also recommend changing "exclusively" to "mainly" and "excluded" to "minimized").

**Response**: We fully agree that the terms "exclusively" and "excluded" were too strong and do not accurately reflect the contributions of plant-derived carbon from root turnover and exudates during the growing season, which are impossible to completely avoid. We also appreciate your point

regarding the presence of some visible residues mentioned elsewhere in the text.

We have revised the manuscript accordingly to more precisely reflect the experimental context and the likely carbon sources:

*Section 2.1 L101-104 (lines 122-123 in the revised version): "...the total inputs to the soil were derived mainly from pig manure"*

*Section 4.1 L286-292 (lines 331-339 in the revised version): "This plant-derived C input is much lower than that from manure, which ranged from 1.6 to 6.4 Mg C ha$^{-1}$ yr$^{-1}$ in the LM and HM treatments, respectively. The rigorous removal of all harvest residues (stalks and recoverable roots) minimized aboveground and root biomass as a source of soil carbon deposition. It should be noted that some residual plant-derived materials (e.g., fine root fragments and exudates) have contributed marginally to the SOC pool. Given the relatively small quantity of plant-derived carbon inputs, the changes in SOC observed in this study are primarily attributable to the application of pig manure."*

In this revision (Section 4.1 lines331-339), we have acknowledged the contribution of plant-derived carbon inputs while emphasizing that the increase in SOC observed in our study was primarily derived from the application of pig manure.

**L128-131: Were the samples shaken with beads?** Either way, this does not typically achieve complete dispersal of all aggregates (e.g., microaggregates < 53 um). I suggest removing the word "completely" and specifying what type/size of aggregate the method was aiming to disperse.

**Response**: Yes, the samples were shaken with beads during the dispersion process. We agree, however, that the term "completely" overstates the dispersion efficiency, as some stable microaggregates often persist. The word "completely" has been removed from the manuscript and we have specified what size of aggregate the method was aiming to disperse in the manuscript:

*Section 2.4 L128-131 (lines 150-153 in the revised version): Briefly, 10 g sieved samples (<2 mm) were dispersed in dilute sodium hexametaphosphate ((NaPO$_3$)$_6$, 0.5%) at a soil: solution ratio of 1:4 by shaking for 18 h (25℃, 180 r min$^{-1}$). This procedure primarily disrupts macroaggregates (>250 μm), but remaining that stable microaggregates (<53 μm).*

**L209: Change "residues" to "residuals"**

**Response**: Revised (lines 238).

**L220: change physic-chemical to physico-chemical**

**Response**: Revised (lines 249).

**L236:** Remove "Despite this" as the finding in this sentence aligns with the larger increase in SOC concentration of POM than MAOM.

**Response**: Removed (lines 269).

**L271-274**: I think it is misleading to say that there was "no correlation between MAOM C and the content of clay and silt" if the R2 was 0.34 and the P value was 0.46. Please revise, e.g., "a weaker correlation".

**Response**: We agree that stating "no correlation" was inaccurate and potentially misleading given the R² value of 0.34. We have revised the sentence to more precisely describe the statistical relationship, changing it to "a weaker correlation" as suggested.

*L 271-274 (lines 310-312 in the revised version): "MAOM C showed no relationship with $Fe_d$ ($R^2 = 0.04$, Fig. 3B), but it was positively correlated with $Fe_o$ ($R^2 = 0.56$, Fig. 3C). A weaker correlation was observed between MAOM C and the content of clay and silt ($R^2 = 0.34$, $P = 0.46$, Fig. 3D)."*

This revision is now fully consistent with the statement in the Discussion (lines 396-L398 in the revised manuscript), which already appropriately describes this relationship as "weak" in contrast to the strong correlation with Feo, thereby providing a coherent interpretation throughout the manuscript.

*Lines396-398 in the revised version: "The strong correlation between MAOM C and non-crystalline Fe oxides ($Fe_o$, $R^2=0.56$; Figs. 3C, 4) highlighted the pivotal role of Feo on SOC stabilization in this red soil."*

L302-309 This section is not clearly written, and the logic is hard to follow. Please revise.

*L302-309 "Recent biomarker evidence demonstrated that plant-derived carbon contributes disproportionately to POM in comparison with MAOM fraction (Zou et al., 2023). Notably, $^{13}C$ isotopic tracing study revealed that 70%-87% of residue-derived SOC was accumulated in the POM fraction at two sites in the study of Mitchell et al. (2021). These results jointly validate that fresh organic inputs are distributed primarily in the POM fraction due to direct occlusion within macroaggregates before microbial processing. Therefore, POM C can be a good indicator of SOC dynamics under field management (Álvaro-Fuentes et al., 2021; Wu et al., 2023)."*

**Response**: We have thoroughly revised this paragraph to improve its logical flow and clarity. The central point we intended to convey is that newly input fresh carbon is preferentially stabilized in the particulate organic matter (POM) fraction. The revised text now builds this argument step-by-step: first stating the principle, then citing supporting biomarker and isotopic evidence, explaining the physical protection mechanism, and finally concluding with the role of POM as an indicator.

*Section 4.1, lines 349-357 in the revised version: "These results implied carbon inputs are preferentially stabilized*

*as POM rather than MAOM. This finding is supported by biomarker evidence showing a great contribution of plant-derived carbon to the POM fraction (Zou et al., 2023). Consistent with this, a $^{13}C$ isotopic tracing study revealed that 70%-87% of residue-derived SOC was accumulated in the POM fraction at two experimental sites (Mitchell et al.,2021). The mechanism behind this preferential sequestration is the physical occlusion of fresh organic material within macroaggregates, which provides initial protection against rapid microbial decomposition. Therefore, POM C can be a good indicator of SOC dynamics under field management (Álvaro-Fuentes et al., 2021; Wu et al., 2023)."*

**L310-311** Similarly, the sentence here is unclear and needs revision (how can POM C be susceptible and simultaneously protected?).

*L310-311: "POM C was susceptible to decomposition due to its rapid dynamics, whereas it was simultaneously protected through physical occultation within soil aggregates."*

**Response**: We agree that the original statement was contradictory and unclear. We have revised the sentence to clarify that the susceptibility of POM C to decomposition is its inherent property due to its chemical nature, but this process can be mitigated through physical protection mechanisms. The revised text now explicitly states that occlusion within aggregates reduces microbial access, thereby slowing decomposition.

*Section 4.1, lines 364-365 in the revised version: "While POM C is inherently labile and susceptible to decomposition, it can be physically protected from rapid decay through occlusion within stable soil aggregates."*

**L315**: Suggest adding "free" before "microaggregates" and editing this paragraph for clarity. As written, this paragraph is confusing because it seems to simultaneously imply that microaggregates decreased (L 315) and increased (L 317). Please clarify.

**Response**: We have revised the paragraph to eliminate the apparent contradiction and improve clarity. As suggested, we have added the term "**free**" before microaggregates to accurately reflect that the measured decrease refers to microaggregates not incorporated into larger structures. The revised text now clarifies that the apparent decrease in free microaggregates is a result of their incorporation into manure-induced macroaggregates, which in turn provides physical protection for SOC.

*Section 4.1 lines 370-376 in the revised version: "In this study, macroaggregates were significantly increased, and free microaggregates were significantly decreased after pig manure application (Table 1). These results implied*

*that manure-derived carbon acted as binding agent, promoting the formation of macroaggregates from free*

*microaggregates. This process provides physical protection for the SOC, particularly that in the POM fraction,*

*within the newly formed aggregates (Peng et al., 2023)."*

**L319**: Change "verified" to "suggest".

**Response**: Revised as suggested (line 376).

**L315**: Is this referring to a difference in percentage mass, or on an absolute basis?? If percentage, it may just be a dilution effect.

**Response**: We incorrectly used the conjunction "while" in this sentence. Our intended meaning was that the observed increase in macroaggregates occurred simultaneously with a decrease in microaggregates, reflecting the conversion of free microaggregates into macroaggregates. Therefore, we have replaced "while" with "and".

**L328**: Higher than what?

*L328: "MAOM retained higher SOC concentration (4.18-7.09 g C kg$^{-1}$ bulk soil) irrespective of treatments…"*

**Response**: The phrase was intended to indicate that the MAOM fraction retained a higher SOC concentration than the POM fraction did. We have revised the sentence in the manuscript to explicitly state the comparison, which now reads:

*Section 4.2 L386-387 in the revised version: "MAOM retained higher SOC concentration (4.18-7.09 g C kg$^{-1}$ bulk soil) than the POM fraction did, irrespective of treatments…"*

**L337**: I think it's misleading to refer to this relationship as "weak" given the R2 is 0.34 and P = 0.046. I suggest "weaker" rather than "weak"

*L337: "The strong correlation between MAOM C and non-crystalline Fe oxides (Fe$_o$, R$^2$=0.56; Figs. 3C, 4)*

*contrasts with its weak relationship with clay + silt content (R$^2$=0.34; Fig. 3D) …"*

**Response**: It has been revised in lines 310-312.

---

## Referee Report (RR1)

**Recommendation to the editor:**

I am pleased to recommend the acceptance of this manuscript in its current form. The authors have done a commendable job in revising the manuscript. They have adequately addressed all the major concerns I raised in the previous round of review. There remain only a few minor errors which can be handled by the editorial office.

**Suggestions for revision or reasons for rejection**

Lines 46: Remove the comma after "Considering".

Lines 64: Use "non-crystalline" instead of "amorphous" for consistency with the terminology used in Lines 57.

Lines 79: Use "MAOM C" instead of "MAOM-C".

Lines 117: Add "applied at 3000 kg Ca(OH)$_2$ ha$^{-1}$ 3yr$^{-1}$" after "lime".

Lines 127: Delete "Lime was applied at 3000 kg Ca(OH)$_2$ ha$^{-1}$ (3a)$^{-1}$" for the HML treatment".

Lines 174-175: Add "%" after "100" in Equation (5) and (6).

Lines 198: Capitalize "control" to "Control".

Lines 241-242: Add the "and relationships between pH and iron oxides" after "the correlations between SOC and iron oxides, soil aggregation…".

Lines 253: Use "that" instead of "those".

Lines 311: Add "but" before "it was positively correlated with…".

Lines 312: Replace "P" with "*P*"

Lines 318-319: Change "while negative correlation with…" to "whereas a negative correlation was observed with …".

Lines 683: Delete "Fed: crystalline oxides; Feo: non crystalline oxides".

Lines 688: Change "/%" to "(%)" in the Table 2.

---

## Author Response (AR2)

We thank the editor and the reviewer for their positive feedback and for accepting our manuscript for publication. We are delighted to hear that our revisions have addressed the concerns raised. We have carefully addressed the minor points noted by the reviewer in this final version, as detailed below.

Lines 46: Remove the comma after "Considering".
Response: Removed.

Lines 64: Use "non-crystalline" instead of "amorphous" for consistency with the terminology used in Lines 57.
Response: The "amorphous" has been replaced with "non-crystalline".

Lines 79: Use "MAOM C" instead of "MAOM-C".
Response: Corrected.

Lines 117: Add "applied at 3000 kg Ca(OH)$_2$ ha$^{-1}$ (3yr)$^{-1}$" after "lime".
Response: Added.

Lines 127: Delete "Lime was applied at 3000 kg Ca(OH)2 ha-1 (3a)-1" for the HML treatment".
Response: The sentence has been deleted.

Lines 174-175: Add "%" after "100" in Equation (5) and (6).
Response: The "%" has been added.

Lines 198: Capitalize "control" to "Control".
Response: Corrected.

Lines 241-242: Add the "and relationships between pH and iron oxides" after "the correlations between SOC and iron oxides, soil aggregation…".
Response: The description has been added.

Lines 253: Use "that" instead of "those".
Response: Replaced.

Lines 311: Add "but" before "it was positively correlated with…".
Response: Added.

Lines 312: Replace "P" with "$P$"
Response: Changed.

Lines 318-319: Change "while negative correlation with…" to "whereas a negative correlation was observed with …".
Response: Modified.

Lines 683: Delete "Fed: crystalline oxides; Feo: non crystalline oxides".
Response: The specified text has been deleted.

Lines 688: Change "/%" to "(%)" in the Table 2.
Response: Changed.